# How Does a Family Caregiver’s Sense of Role Loss Impact the Caregiving Experience?

**DOI:** 10.3390/healthcare9101337

**Published:** 2021-10-08

**Authors:** Lisa Ribeiro, Bach Q. Ho, Dai Senoo

**Affiliations:** Department of Industrial Engineering and Economics, School of Engineering, Tokyo Institute of Technology, 2 Chome-12-1 Ookayama, Meguro City, Tokyo 152-8550, Japan; ho.b.aa@m.titech.ac.jp (B.Q.H.); senoo.d.aa@m.titech.ac.jp (D.S.)

**Keywords:** family caregiver, sense of role loss, role exit, role transition, role rotation, sense of personal choice in life and self-priority

## Abstract

Family caregivers reportedly have a powerful sense of role loss, which is felt when one senses a change in role or responsibility, relationship distancing, or a changed asymmetry. Little is known about the impact it has on the caregiving experience, so the purpose of this study was to clarify this in three distinct settings: when an individual’s primary role changed to the caregiver role after the start of caregiving; when their primary role was other than the caregiver role after this start; and when their primary role was the caregiver role before caregiving started. Sixty-six individuals responded to an online survey, and a framework method was employed to organize the collected data and uncover themes for analysis. Our findings shed light on the sense of caregiver role loss and pointed to the possibility of generating it when family caregivers rotate through held roles and the use of it as a tool to maintain or regain a sense of personal choice in life and self-priority. Our study is probably the first to analyze this phenomenon in different caregiving settings based on an individual’s primary role and role transitions and brings to light a new perspective of the phenomenon by understanding how it arises, its nuances, and its impact on the caregiver’s experience.

## 1. Introduction

Caregiving is an activity directed at a person with a disabling condition or illness unable to care for themselves and with three main objectives: to get things done, to accomplish a sense of health and well-being for the care recipient, and to accomplish a sense of health and well-being for the caregiver [1]. The outcomes associated with the activity vary from positive to negative, such as life satisfaction, personal growth through learning opportunities [2], caregiving stress, role captivity [3], objective and subjective caregiver burden [4], less time for other activities, financial pressure, and tiredness [5].

Family caregivers are unpaid workers who produce health and may be considered hidden patients of the care recipient’s physician [6]. They are members of the healthcare team caring closest to the care recipient and members of the unit of care, and their concerns, needs, and health must be paid attention to and addressed [7]. These caregivers are essential to healthcare systems. For instance, the American Association of Retired Persons (AARP) estimates that in 2017 alone, they delivered 34 million hours of care with an estimated unpaid contribution of 470 billion dollars only in the USA [8].

Throughout the caring process, family caregivers reportedly have a powerful sense of role loss felt when one senses a change in role or responsibility, relationship distancing, or a changed asymmetry [9]. Little is known about how this sense of role loss impacts these individuals’ caregiving experiences.

To build tailored solutions that will effectively support family caregivers and account for the uniqueness of each caregiving experience [10], it is necessary to develop a deeper understanding of the phenomenon of sense of role loss, its triggers, the impact it has on the caregiver’s caregiving experience, how this impact differs among caregiving settings and, from each setting, uncover mechanisms deployed to deal with it.

The objective of this study is to understand the impact of a sense of role loss by contrasting the caregiving experience of family caregivers in different groups, where each group configures a different caregiving setting. One of its contributions is bringing to light a new perspective of a sense of role loss that can work as a tool with implications on the caregiver’s sense of personal choice and self-priority. Furthermore, we researched this phenomenon based on the primary role and role transition and then contrasted the experiences. Prior studies account for caregivers’ experiences based on identity, condition, illness, and gender, so we can contribute to this list with these additional settings.

## 2. Theoretical Background

Sense of role loss resembles what Ebaugh identified as “the vacuum” experienced by those who go through a voluntary role exit process [11] (p. 143) and find themselves in a state of being roleless [12] (p. 136). A role may be defined as a set of expected positional behaviors in a social group which includes a social position with normative expectations to be enacted through the behavior of the individual occupying that position [13]. These patterns of behaviors are learned, and role performance encompasses tasks and skills [14]. In this study, a primary role is either the role most central to an individual’s identity [15] or the role one performs more frequently. In a subjective ranking system, individuals have primary, secondary, tertiary roles, and so on, contributing to the shaping of their identity in differing degrees. 

The family caregiver role emerges from an existing role relationship and should be seen as this role relationship’s transformation, holding strong meaning to those performing in it and regulated by norms or social rules [10]. For instance, a husband–caregiver role can be seen as a transformation of a husband role.

According to Zena Blau, a role exit occurs “whenever any stable pattern of interaction and shared activities between two or more persons ceases” [11] (pp. 13, 14) and is defined as a “process of disengagement from a role that is central to one’s self-identity and the reestablishment of an identity in a new role that takes into account one’s ex-role” [11] (p. 1). It is suggested that the extent to which the exit is problematic depends on how it occurred and the options available for new, valued roles after it [16] (p. 1). Caregivers may fully exit a prior role and create the “ex-role” or not, but a sense of role loss is a reported experience.

It is not felt exclusively by family caregivers and, as people transition through roles, their loss and gain happen constantly [17]. Within the same role, the required performance changes over time as one goes through both planned and unplanned life transitions [18], which may result in macro or micro role transitions. 

Macro role transitions occur when an individual moves between sequentially held roles, are less frequent, and may involve more or less permanent changes [12] (p. 7): one disengages from a role (role exit) and engages in another (role entry) (e.g., placement of the care recipient in a nursing home and the ceasing of the full care activity by the caregiver), being lengthy and demanding. Regularly, individuals go through micro role transitions: they psychologically (and, sometimes, physically) move through simultaneously held roles in short periods, comprising of temporary and recurrent role exits and entries [12] (p. 261). One holds multiple roles and, depending on the context the individual finds him/herself in, a specific role is occupied while others remain in the background.

Table 1 consolidates studies that highlighted the experience of a sense of role loss when life transitions occurred.

There is yet a lot to be uncovered about the family caregiver’s sense of role loss, its impacts on the caregiving experience, and how these individuals deal with it. In the caregiving context, role changes within the family can occur quickly, may extend for a short or long time, or can be permanent, resulting in caregivers finding it difficult to adjust to the impact of the care recipient’s illness. Some individuals experience role loss, role gains, taking in the identity, or viewing it as an extension of an existing role [24] (p. 122).

Although each caregiving situation is always unique [10], similarities in caregiver roles indicate that (i) the majority are female family members; (ii) the activity changes over time; (iii) caregiving leads to changes in the relationships and identities of both the caregiver and the care recipient; (iv) caregiving is accompanied by stress and burden impacting the caregiver’s well-being; and (v) there are positive outcomes such as self-satisfaction and sense of mastery [25] (pp. 177–182).

Despite these similarities, the caregiving experience varies widely across groups and within the same group of caregivers [10,26,27]. It is associated with caregiving esteem and schedule burden [28] and with the degree of involvement in personal care increasing emotional and psychological burden [29]. Although caregiving is perceived as rewarding and a positive experience is associated with a caregiver’s better mental health, it reduces the caregiver’s time for recreation or their own activities [30]. 

Many studies shed light on the family caregivers’ experience per identity, condition, illness, or gender. However, it remains unclear whether the primary role was the caregiver role and how this variability in the caregiving setting was associated with the reported sense of role loss. Furthermore, little is known about the impact of sense of role loss on the caregiving experience, its triggers, how it is expressed or manifested, and how it is dealt with, as most studies stopped at reporting the phenomenon with brief mentions of mitigation strategies. In an attempt to fill this gap, we formulated the following research question: 

How does a family caregiver’s sense of role loss impact the caregiving experience?

## 3. Materials and Methods

### 3.1. Sample

We employed the framework method of qualitative data analysis to organize the data and uncover themes [31] concerning the sense of role loss and its impact on the caregiving experience. To create a large and standardized sample of data [32] (p. 361) without causing much inconvenience to potential respondents, empirical data were collected through an online survey with Google Forms [33] (p. 57), as detailed in Table 2. The overall goal of the survey was to capture the experience of the family caregivers through their reflections about the role, their actions under the light of the role transition they went through, and the perceived role changes to uncover how a sense of role loss impacted their experience.

We reached out to family caregivers’ associations through their webpages’ contact forms and to caregivers who the researchers were acquainted with over e-mail or Whatsapp message. A total of 66 individuals responded to the questionnaire. Based on their answers about their primary roles before and after the start of the caregiving activity, three distinct groups of respondents emerged and are presented in Table 3.

At first, we would contrast the caregiving experience of Groups 35 and 27 only, because there was a clearer point in time when the caregiving activity started. Thus, we assumed that members of these groups went through role transition(s) and the family caregiver’s sense of role loss occurred. However, leaving Group 4 out of the analysis would be unfair to its members as well as a lost opportunity for us to understand these individuals’ role transitions and the sense of role loss they felt, thus enriching the analysis.

The demographics of each group are displayed in Table 4. 

The majority of respondents were women performing in the role for more than five years. In Group 35 and 27, most were led into caregiving due to the care recipient’s illness; in Group 4, most were led into caregiving due to the care recipient’s injury. The majority shared the same home with the care recipient. Respondents in Groups 35 and 4 who shared tasks with others corresponded to 40% and 25%, respectively, whereas in Group 27, this percentage jumped to 59%.

### 3.2. Data Analysis

The demographics served to group the respondents and to paint an overall picture of each group. Next, we organized the answers to open questions in Section 4: the ERA (Experience–Reflection–Action cycle) [35] (pp. 44, 45) and the Reflective Cycle [36] (pp. 49, 50) frameworks. ERA was chosen because it was developed to enable reflective practice in healthcare and, in this study, family caregivers were conceptualized as practitioners [1] and members of the care team [7]. We employed the Reflective Cycle to break down the analysis of each ERA component and objectively sort the data.

Developed for the nursing practice, the ERA framework helps these professionals constantly reflect on their experience and act based on new perspectives generated from this reflection [37] (p. 2). It looks at experience, reflection, and action cyclically where one component feeds the next, and it allows individuals to use it as a process tool instead of a one-timer. ERA is suitable to analyze the respondents’ answers about their caregiving experience because they are practitioners who reflect on their experiences and act to accomplish goals.

The component “Experience” is what one lives through, “things that happen to me”. “Reflection” is looking back at what happened to generate new understandings and perspectives. This should result in an “Action”, the application in the practice of what was learned during reflection [35] (pp. 44, 45). We assumed that the experiences shared by the respondents stood out to them in some way [38] (p. 38) and therefore were fit for analysis with the framework.

To guide the data sorting, each ERA component was paired with the phases of Gibbs’ Reflective Cycle [35] (p. 61) to achieve granularity in the thematic analysis by posing objective questions to be answered with the data. Throughout the cycle, an individual describes the experience, verbalizes feelings and thoughts about it, evaluates it, analyzes to make sense of it, concludes a learning point or how the experience should have been dealt with, and, lastly, devises an action plan for future similar situations. Each Reflective Cycle phase was answered with their corresponding questions in the survey, as illustrated in Figure 1.

The length of the answers to the open-ended questions varied from one word to long, reflective paragraphs. As illustrated in Figure 1, these answers went through a two-phase sorting process: Phase I—Reflective Cycle (description, feelings, evaluation, analysis, conclusion, and action plan) and survey questions;Round 1—Answers that conveyed the same theme were grouped. Themes emerged from this grouping.Round 2—The total number of answers under each theme was summed up, resulting in a score.Round 3—The themes were ranked from highest to lowest score. Please refer to Appendix A for the list.Phase II—Reflective Cycle and ERA (experience, reflection, and action);Round 4—Based on the results of Part I, themes from description, feelings, and evaluation were combined in experience; themes from analysis and conclusion were combined in reflection; themes from the action plan were carried to action.Round 5—The same themes were combined into one, and their scores were summed up.Round 6—The top three themes with the highest scores were ranked and selected in the ERA framework, as displayed in Table 5.

## 4. Results

### 4.1. Experience

The theme with the highest score in Group 35′s answers reported a need to focus on the present situation accepting one cannot know what comes next (i.e., TH063). In Group 27, a similar theme ranked third, and it was phrased as a need to be constantly alert, monitor the situation, and adapt to it (i.e., TH019), but it did not appear in the answers of Group 4. So, the first finding of this study was that caregiving required that caregivers have a heightened focus on the present circumstance to respond fast to the intense, constantly changing demands of caregiving, and this varied among the three settings. Family caregivers found themselves devoting different amounts of their time to support care recipients the best way they can. Time, or the lack of it, was a recurrent theme in their answers.

The necessity to focus one’s attention on a current circumstance impacted the time left for activities outside the caregiver role. This altered the relationship that family caregivers had with the past, present, and future times. The alteration of relationship with present and future times was clear in the groups; the respondents in Group 27 reported their sense of unpreparedness to provide care to the care recipient, with some mentioning a worry (i.e., TH003) that led to insecurity about what to do now and what the future would hold for both the caregiver and the care recipient, as explained by a respondent:


*“Despair. As for the future of my son and mine. I was still trying to assert myself professionally, I had a double shift—work and study”.*

*(Caregiver 005)*


The same theme was found in Group 35 (i.e., TH051) and Group 4 (i.e., TH101), producing similar feelings in respondents, altering the relationship these respondents had with their present and future times. Additionally, these family caregivers had the relationship with their past time altered, as they saw a portion of activities they used to carry out or roles they used to perform as no longer a part of their present lives. They reported little time to pursue personal and professional lives (i.e., TH052); low priority on personal life activities and difficulty planning activities with others (i.e., TH065); and the need to put aside parts of their life to assist other’s needs and be an extension of the care recipient (i.e., TH104). 

The need for heightened focus on the present altered the relationship that the respondents of the three groups had with their present and future times. Family caregivers in Groups 35 and 4 indicated that the relationship with their past time was altered after the start of the activity.

### 4.2. Reflection

When reflecting on the meaning of the caregiver role and the process of settling in it, all groups stated how crucial their role was in the care recipient’s life, and they perceived caregiving as work with tasks and goals (i.e., TH070, TH071, TH087, TH024, TH109, TH110) that required being present, availability, dedication, and alertness. It was also about giving love (i.e., TH073, TH027, TH111).

Responses from Groups 35 that revealed changes to respondents’ personal lives made it to the top three themes, be it because they felt they gave up personal choice in life or gave up on activities of personal life, in both cases to assist others with their needs and favor their care, as one family caregiver put it:


*“Caring without schedules, stop having freedom and time for me.”*

*(Caregiver 019)*


Group 35 reported they had little chance to pursue personal and professional lives activities due to a lack of time, because of the required availability or the need to prioritize caregiving. The altered relationship with the past time gave rise to a sense of role loss expressed through reflections such as “I stopped having a social life” (Caregiver 007) where the caregiver used to perform in and rotated through roles no longer held, impacting their caregiving experience by contributing to a diminished sense of personal choice and self-priority to assist others with their needs, which is defined in this article as the ability to practice self-priority and pursue one’s life activities outside the caregiver role while holding this role. 

Such a theme did not make it to Group 27′s top three themes, but it did appear in Group 4′s. However, the difference from Group 35 is that the theme of “not having a life” was expressed in comparison to others “having a life”, and we interpreted it as a wish to, from time to time, step out of caregiving, rotate through held roles, and perform in them as others appeared to be doing, as Caregiver 15 put it:


*“Being the youngest, I have always lived with her and took care of her. First, I played with her and, naturally, gave her meals, changed her diaper, and, when I was strong, dressed and bathed her, made her meals so it was easy. Nowadays it is more difficult... I see others having a life and I don’t.”*

*(Caregiver 015)*


In Group 27, the respondents either continued to perform in the same primary role as before the activity started or they changed primary roles. In both cases, caregiving was a role they took on or an added feature to an already held one. These individuals continued to rotate through roles, they exited and entered them regularly and experienced a sense of role loss, particularly a sense of caregiver role loss. 

The possibility to rotate more frequently through roles allowed the family caregivers to step out of caregiving from time to time and generate a sense of caregiver role loss that can work as a tool used to maintain or regain a sense of personal choice in life and self-priority. In this study, role rotation was defined as the deliberate act of exiting and entering held roles, occupying, and performing in the salient role of the context, and generating a sense of role loss from the role one exited.

### 4.3. Action

Learning, knowledge, and understanding of the care recipient and the condition was the common theme to Groups 35 and 27 (i.e., TH090, TH041). Responses indicated that the caregiving demands might change fast. All three groups reported the effectiveness of support received from professionals, associations, and attendance to training to access knowledge, develop skills, reduce the sense of unpreparedness, improve the quality of the care they provided, and help them balance these tasks with the pursuit of personal life activities (i.e., TH091, TH043, TH118, TH120), as Caregiver 024 put it:


*“In the early years, it was very difficult for me, especially on a psychological/emotional level. Over the years and thanks to the doctors who accompany my mother, I have been able to ‘digest’ or deal better with the whole situation. I am aware that future times will become more difficult and complicated, but I will continue to care and love unconditionally.”*

*(Caregiver 024)*


In Group 35, caregiving was viewed as a constantly changing process that required continuous learning and focus on the current situation for adaptation (i.e., TH090). This was found in Group 27, which was phrased as understanding the situation, having the determination to overcome challenges that appeared along the way, and a realization that the role required learning (i.e., TH044, TH042, TH041). On the other hand, a similar theme did not appear in any of the responses of Group 4.

The actions of Groups 35, 27, and 4 pointed to knowledge and learning for suitable adaptation to and performance in the caregiver role. Groups 35 and 27 indicated a need to focus on the present situation. Group 4 revealed the maintenance/participation in activities/events outside the role (i.e., TH117) and a need to leave the job to dedicate to care (i.e., TH119). 

The actions to settle in the caregiver role aimed at developing ways to deal with the necessity of heightened focus on the present situation and mitigate its effect on the relationship family caregivers had with their past, present, and future times. It enabled them to continuously construct the role to either cope with a sense of role loss or generate it to work as a tool to help them pursue a balance between their role as a caregiver, work, and their personal life.

### 4.4. Network of Themes

Based on the previous findings, we deepened our analysis of the groups by gaining visibility on how their caregiving experiences related to one another through a network of themes in each ERA component. The same themes across groups were connected by a continuous line that represented a strong relationship (1 point). Similar themes with their nuances were connected by a dashed line and considered a weak relationship (0.5 point).

Common to all three groups was a sense of unpreparedness, as illustrated in Figure 2. Groups 35 and 4 shared the effect caregiving had on the pursuit of their personal lives’ activities. The need to be alert and monitor the situation loosely connected Groups 35 and 27.

The themes in reflection composed the densest network of all, as Figure 3 shows. It revealed a myriad of interpretations with strong and weak connections between the groups whose members perceived caregiving as work in which the focus and priority was the care recipient. All three groups also saw the caregiver role as either an act of love or to give love and support to the care recipient. Groups 35 and 4 related when they verbally expressed the effect caregiving had on their personal lives’ activities and how it demanded availability.

Caregiving is far from an easy activity. Settling in the role and managing the changes it brings into the lives of those involved requires various strategies suitable to the various experiences. The network of themes in action showed the three groups acknowledged the positive impact of the support received from professionals (i.e., healthcare, housekeeper), associations, attendance to training to acquire skills and build their confidence, as illustrated in Figure 4. Groups 34 and 27 shared the necessity of continuously learning as caregiving progressed and its demands changed.

Based on the three networks, it was possible to calculate the strength of relationships among the experiences of the three groups based on the scores of continuous and dashed lines that connected the groups. With the scores, we were able to view how strong each group’s experience related to the other, as displayed in Figure 5.

Groups 35 and 4 scored the highest number of the same or similar themes in their experience: it was surprising to see that a sense of role loss was felt by these two groups and impacted their experience, because we expected Group 4′s transition into the caregiver role to be the most gradual, since these individuals reported they were caregivers before the activity had started. These two groups expressed the need to label the pursuit of personal life activities as a lower priority than assisting the care recipient and attending to caregiving demands, which impacted their sense of lack of personal choice in life and self-priority. Group 4 also mentioned it comparatively (e.g., “difficult to see others living their lives while they are not”).

The relation of experiences between Groups 35 and 27 had the second-highest score, whereas the weakest relationship was between Groups 27 and 4. Given these results, we could see that Group 27’s experience related the least to other groups, and this led us to think that the sense of role loss impacted their caregiving experience differently. These findings are discussed next.

## 5. Discussion

This study aimed at understanding how the caregiver’s sense of role loss impacted the caregiving experience of “primary role transformed” caregivers (those in Group 35 and whose primary role was the caregiver role after caregiving started); “primary role preserved” caregivers (members of Group 27 and whose primary role was other than the caregiver role after the start of the activity); and “constant” caregivers (respondents in Group 4 and whose primary role was the caregiver role before caregiving started). Addressing this variability in the caregiving settings allowed us to shed light on how the sense of role loss differed per group and, consequently, how it impacted the experience.

The first finding was that caregiving required that caregivers had a heightened focus on the present situation for a quick response to its changing demands and varied among the groups. This altered the relationship the respondents had with their present and future times—and in the case of primary-role-transformed and constant caregivers, their past time, too. Uncertainty for the future [9,39] and an inability to plan [39] or live in the present were recurrent themes for these individuals. An imbalance between caregiving, the caregiver’s personal life, and efforts to achieve a balance between the two [40] was expressed as “no time for personal life” and did not make it to the top recurrent themes of primary-role-preserved caregivers.

The altered relationship with the past time and roles experienced by those with a primary role transformed gave rise to a perception of a life before caregiving that could not be lived at that moment. Constant caregivers expressed similar feelings and compared them with other people who were “having a life”. The necessity to be available and of alertness to the current situation hindered the caregiver’s ability to plan the pursuit of personal and professional activities they used to carry out or wished they did. For them, entering the caregiver role resembled a macro role transition with a permanent exit from roles due to a life transition. Primary-role-preserved caregivers took on the caregiver role and either maintained their primary role or changed it to a role other than caregiver; they kept going through micro role transitions by rotating roles and performing in them.

The second result showed that concerning primary-role-transformed and constant caregivers, the altered relationship with the past time and the feeling of permanent exit from past roles gave rise to a sense of role loss that impacted their caregiving experience by contributing to a diminished sense of personal choice and self-priority, as they prioritized the assistance to others with their needs and perceived they were no longer able to perform in past roles. Prior studies found that those who felt they did not have a choice about entering the caregiver role were three times more likely to report stress [41], with perceived lack of choice associated with higher emotional stress, physical strain, and negative impacts on health [42], whereas a sense of free choice in entering the role was positively associated with the family caregiver’s well-being [43]. 

These studies focused on the choice of entry in the role before caregiving and the caregiver’s well-being during the activity, not on the choice of entry in roles other than the caregiver role while carrying the activity out. In addition, it is not uncommon that individuals have little or no choice but to become family caregivers, so the concept of choice during the activity may take different shapes that were not yet clear. Our study contributes to advance this knowledge by understanding the dynamic interplay among role rotation, sense of (caregiver) role loss, and sense of personal choice in life and self-priority during caregiving, not before it.

Primary-role-preserved caregivers took on this role or added its features to an already held one. These individuals carried on with other roles, transited through them, and experienced a sense of role loss, but in their reports, it did not appear to have generated a diminished sense of personal choice in life and self-priority. Research shows that family caregivers performing in the marital and the employment roles reported better psychological and physical health [44], and those caring for someone with a mental disability experienced fewer stress outcomes as they spent more hours in outside work [45]. We concluded that these individuals experienced a sense of caregiver role loss by stepping out of caregiving from time to time.

We argue that the possibility to hold and rotate through different roles helps caregivers expand capabilities to promote activities and lifestyles they value and flourish within the circumstances at hand [46] (location no. 444, 453). Role rotation helps generate a sense of caregiver role loss that can be used as a tool to equalize family caregivers’ freedom of functioning [47] (p. 139) and maintain a sense of personal choice in life and self-priority while performing in this role as well.

Third, focus on the current situation and possession of knowledge and skills were mechanisms employed by family caregivers to deal with what was at hand; professionals’ and associations’ support facilitated their acquisition. This finding is aligned with prior studies that indicated that preparedness was associated with caregivers’ higher levels of hope and reward, lower levels of anxiety [48], and the need for knowledge and learning [49], adding that preparedness helped caregivers maintain or regain a sense of control over their relationships with the past, present, and future times by generating a context where they may create new meanings and narratives about their own care experience.

Through the findings, we were able to understand the interplay of the sense of caregiver role loss with two other concepts that emerged in the study: role rotation and sense of personal choice in life and self-priority, as illustrated in Figure 6.

This relationship helped us understand why the experience of primary-role-transformed and constant caregivers related more often: according to their answers, they had fewer possibilities to rotate through held roles, step out of caregiving, generate a sense of caregiver role loss, and increase or regain a sense of personal choice in life and self-priority. Primary-role-preserved caregivers took on this role, kept rotating through others, and generated a sense of caregiver role loss that contributed to the maintenance of their sense of personal choice in life and self-priority.

Practical implications are presented to family caregivers and the support system (i.e., healthcare professionals, enterprises, associations, governments). They are consolidated in Table 6 and Table 7 respectively and contain the excerpts of family caregivers’ responses so we can hear from them how they managed their caregiving experience.

On a theoretical level, this research contributes to the knowledge about the caregiving activity and the impact of the sense of role loss on the caregiver’s experience. Prior studies identified this feeling, portrayed contexts where it was felt, and defined what it is. Our study delved into the caregiver’s sense of role loss, uncovered its impact on the caregiving experience, shed light on the sense of caregiver role loss, and pointed to the possibility of generating it when these individuals rotate through held roles. Role rotation allows family caregivers to step out of caregiving from time to time and the sense of caregiver role loss to be used as a tool to maintain or regain a sense of personal choice in life and self-priority. 

Most studies stopped at reporting this phenomenon, and very few mentioned mitigation strategies to negative effects. Our motivation to study it more deeply was to understand how it affects caregivers and better support them since a sense of role loss seems unavoidable. Additionally, we studied it under three different caregiving settings based on their primary role and role transition; then, we contrasted the experience of primary-role-transformed, primary-role-maintained, and constant caregivers. Prior studies account for identity, condition, illness, and gender, so we can contribute to this list with these settings. The motivation behind looking at different settings was to account for variability, have different perspectives to look from, and see the multiple faces of the sense of role loss.

One limitation of this study is the sample size of constant caregivers (Group 4), and, despite respondents’ rich input, we believe more answers would have led us to more insights about their experience. The data analysis did not consider the caregiving duration, gender, sharing of tasks, and how they contribute to turning sense of role loss into a tool. Furthermore, this study looked at the sense of role loss as a one-time-only phenomenon, but we could see it is multi-faceted and happens multiple times as care demands change, so it will be useful to understand its stages and characteristics. We recommend future studies to tackle these limitations to help us advance our knowledge about caregiving and effectively support a wider range of caregivers.

Caregiving is rarely easy or smooth. Family caregivers are a unique type of practitioner because the role emerges from existing relationships [10]. Understanding the phenomenon of the sense of role loss deeply can uncover new ways of improving the caregiver’s experience and its impact on caregiving burden, stress, quality of care, and overall well-being of caregivers and care recipients.

## 6. Conclusions

This article’s goal was to understand how a family caregiver’s sense of role loss impacts the caregiving experience of family caregivers. The data for analysis were collected through an online survey and answered by sixty-six individuals. The framework method was employed to organize the data into themes for analysis. Our findings shed light on the sense of caregiver role loss and pointed to the possibility of generating it when these individuals rotate through held roles. Role rotation allows family caregivers to step out of caregiving from time to time, and the sense of caregiver role loss can be used as a tool to maintain or regain a sense of personal choice in life and self-priority.

## Figures and Tables

**Figure 1 healthcare-09-01337-f001:**
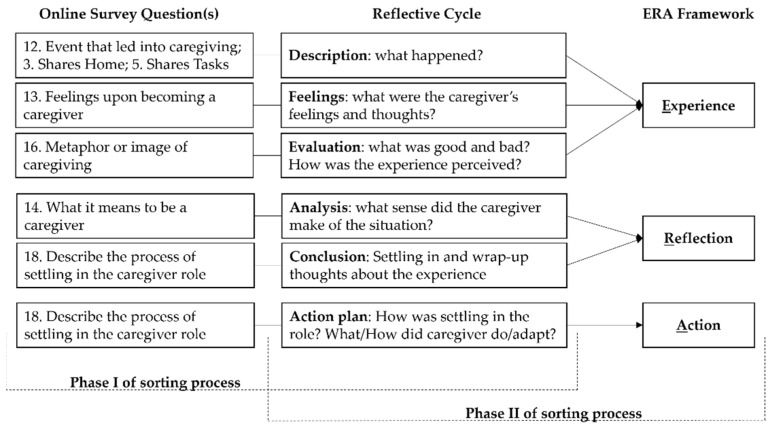
ERA, Reflective Cycle, and survey questions’ data-sorting process.

**Figure 2 healthcare-09-01337-f002:**
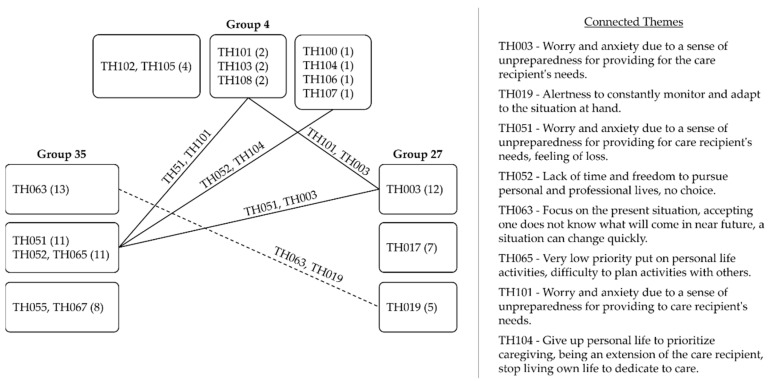
Network of themes in experience.

**Figure 3 healthcare-09-01337-f003:**
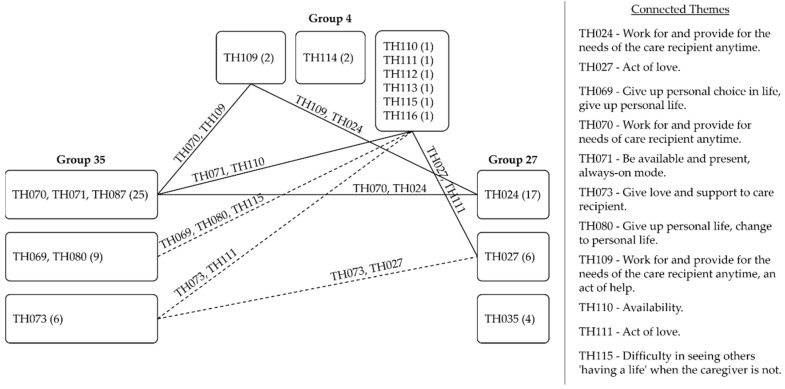
Network of themes in reflection.

**Figure 4 healthcare-09-01337-f004:**
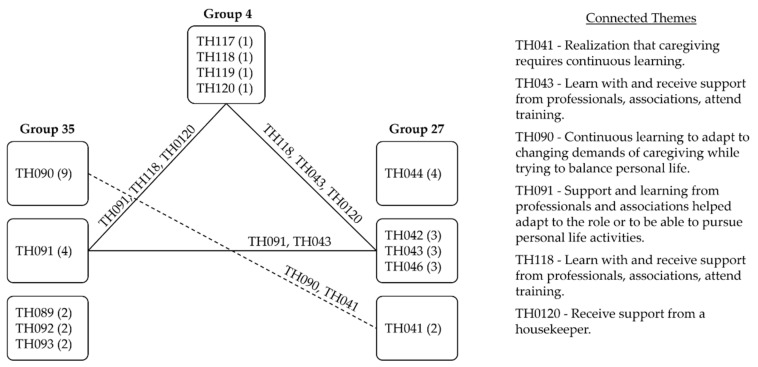
Network of themes in action.

**Figure 5 healthcare-09-01337-f005:**
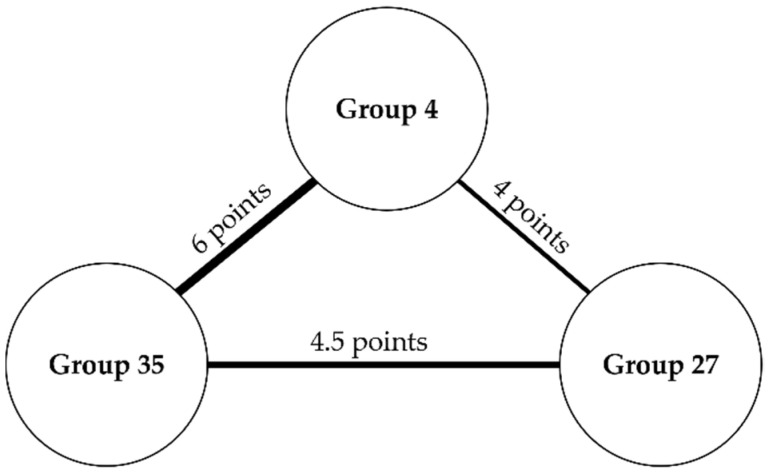
Diagram of groups’ experiences and strength of relationships.

**Figure 6 healthcare-09-01337-f006:**
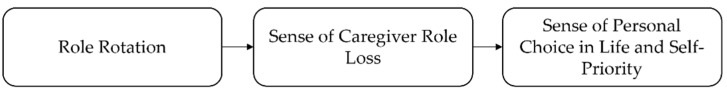
Role rotation, sense of caregiver role loss, and sense of personal choice in Life and Self-Priority.

**Table 1 healthcare-09-01337-t001:** Studies on the sense of role loss.

Study	When It Is Felt	by Whom	Cause/s	Mitigation
Hasselkus [9]	Throughout caregiving.	Caregiver	Change in role and responsibility; Relationship distancing or a changed asymmetry.	
Dellasega et al. [19]	After the care recipient’s placement in a nursing home.	Caregiver	Nursing home placement.	Redefinition of the caregiver role (expressed longing to resume prior roles).
Kellett [20]	After the care recipient’s placement in a nursing home.	Caregiver	Sensed change in engaged involvement.	Finding new ways of caring for the relative.
Kim and Moen [21]	Upon and after retirement.	Retiree	Exiting role central to identity and environmental loss accompanying the role exit.	
Williams [22] (p. 106)	Foster child arrival at or departure from the family.	Birth children	Place or position constant re-alignment to accommodate a foster child.	
Tummala-Narra and Deshpande [23] (p. 175)	Post-immigration.	Immigrant	When shifts in power, respect, and authority within the families occur.	

**Table 2 healthcare-09-01337-t002:** Online questionnaire details.

Section	Title	Contents	Objective
0	Caregiver and sense of role loss	A brief explanation about the research, clarification of concepts, and collection of informed consent.	To ensure the respondent was aware of the questionnaire’s purpose in the context of this study.
1	Demographics	Information about gender, duration of caregiving, if the home was shared with the care recipient, if tasks were shared with others, and relationship with the care recipient.	To compose a picture of the caregiving context.
2	Before caregiving activity started	Respondent to indicate primary, secondary, and tertiary roles (the list of roles presented to the respondents was based on “The Role Checklist” [34] designed to elicit information about an individual’s occupational roles, which are: student; worker; volunteer; caregiver; home maintainer; friend; family member; religious participant; hobbyist/amateur; participant in organizations; other) and self-rated performance in each before the start of caregiving activity.	To visualize respondents’ top three main roles and understand how they perceived their performance.
3	After caregiving activity started	Respondent to indicate primary, secondary, and tertiary roles (Ibid.) and self-rated performance in each after the start of caregiving activity.	To visualize respondents’ top three main roles and understand how they perceived their performance. See the change in the roles after caregiving started.
4	The caregiver	Open-ended questions about what event led the respondent into caregiving, feelings upon becoming a caregiver, what it means to be a caregiver, process of settling in the role, a metaphor or image to describe the role, and checklist of tasks.	To give voice to respondents and get a glimpse of their world to better understand their experience.

**Table 3 healthcare-09-01337-t003:** Groups of respondents.

Group	Setting	Name and Description	Total
35	The primary role is ‘caregiver’ after the start of caregiving	Those whose primary role changed to the caregiver role after caregiving started.	35
27	The primary is other than ‘caregiver’ after the start of caregiving	Those whose primary role did not change to the caregiver role after caregiving started: they either preserved their previous primary role or changed to a role other than caregiver.	27
4	The primary role was ‘caregiver’ before the start of caregiving	Those whose primary role was the caregiver role before caregiving started.	4

**Table 4 healthcare-09-01337-t004:** Demographics of Groups 35, 27, and 4.

Group	Event	Gender	Duration of Caregiving	Shares Home	Shares Tasks
35	Illness—20 (60%) Injury—7 (20%) Death of in-charge of care—4 (10%) Disability—4 (10%)	Female—28 (80%) Male—7 (20%)	More than 5 years—22 (63%) 2 to 3 years—4 (11.4%) 3 to 4 years—4 (11.4) 4 to 5 years—2 (5.7%) Less than 1 year—1 (2.8%) No longer caregiver—2 (5.7%)	Yes—29 (83%) No—6 (17%)	Yes—14 (40%) No—21 (60%)
27	Illness—15 (56%) Death of in-charge of care—7 (26%) Disability—2 (7%) Injury—2 (7%) Help—1 (4%)	Female—20 (74%) Male—7 (26%)	More than 5 years—18 (66.7%) 3 to 4 years—4 (14.8%) 4 to 5 years—2 (7.4%) 1 to 2 years—1 (3.7%) Less than 1 year—1 (3.7%) No longer caregiver—1 (3.7%)	Yes—23 (85%) No—4 (15%)	Yes—16 (59%) No—11 (41%)
4	Injury—2 (50%) Disability—1 (25%) Illness—1 (25%)	Female—4 (100%)	More than 5 years—4 (100%)	Yes—4 (100%)	Yes—1 (25%) No—3 (75%)

**Table 5 healthcare-09-01337-t005:** ERA and the caregiver’s experience—consolidated themes and scores.

Group	Experience Theme (Score)	Reflection Theme (Score)	Action Theme (Score)
35	TH063—Focus on the present situation, accepting one does not know what will come in near future, a situation can change quickly. (13)	TH070—Work for and provide for needs of care recipient anytime; TH071—Be available and present, always-on mode; TH087—Care recipient is the priority, dedication. (25)	TH090—Continuous learning to adapt to changing demands of caregiving while trying to balance personal life. (9)
	TH051—Worry and anxiety due to a sense of unpreparedness for providing for care recipient’s needs, feeling of loss. (11) TH052—Lack of time and freedom to pursue personal and professional lives, no choice; TH065—Very low priority put on personal life activities, difficulty planning activities with others. (11)	TH069—Give up personal choice in life, give up personal life; TH080—Give up personal life, change to personal life. (9)	TH091—Support and learning from professionals and associations helped adapt to the role or to be able to pursue personal life activities. (4)
	TH055—Responsibility, duty; TH067—Responsibility, duty. (8)	TH073—Give love and support to the care recipient. (6)	TH089—Focus on the current situation and attend to it as it requires. (2) TH092—Responsibility, duty. (2) TH093—Care became a routine. (2)
27	TH003—Worry and anxiety due to a sense of unpreparedness for providing for the care recipient’s needs. (12)	TH024—Work for and provide for the needs of the care recipient anytime. (17)	TH044—Understanding of the care recipient and their condition. (4)
	TH017—Be present to support and understand the care recipient. (7)	TH027—Act of love. (6)	TH042—Determination to face and overcome challenges. (3) TH043—Learn with and receive support from professionals, associations, attend training. (3) TH046—Immediate and instinctive settling in the caregiver role and its demands. (3)
	TH019—Alertness to constantly monitor and adapt to the situation at hand. (5)	TH035—Difficulty in settling in the caregiver role. (4)	TH041—Realization that caregiving requires continuous learning. (2)
4	H102—Lost, alone, lack of support; TH105—Lack of support, process inefficiencies, paperwork. (4)	TH109—Work for and provide for the needs of the care recipient anytime, an act of help. (2) TH114—In the past, easier. Now, more difficult, less energy than when started. (2)	TH117—Maintenance of/participation in activities/events outside the caregiver role. (1) TH118—Learn with and receive support from professionals, associations, attend training. (1) TH119—Left job to focus on care. (1) TH120—Receive support from a housekeeper. (1)
	TH101—Worry and anxiety due to a sense of unpreparedness for providing for the care recipient’s needs. (2) TH103—Courage, gratitude, strength from love. (2) TH108—Unexpected turn of events, a sudden new reality to attend to. (2)	TH110—Availability. (1) TH111—Act of love. (1) TH112—Comforting. (1) TH113—Exhaustion. (1) TH115—Difficulty in seeing others ‘having a life’ when the caregiver is not. (1) TH116—Had to do what was necessary to support the care recipient/s. (1)	
	TH100—Caregiving as a natural/anticipated path. (1) TH104—Give up personal life to prioritize caregiving, being an extension of the care recipient, stop living own life to dedicate to care. (1) TH106—Resources demanding activity (especially, financially). (1) TH107—Love. (1)		

Based on these themes, the results will be presented in the following section.

**Table 6 healthcare-09-01337-t006:** Practical implications to the family caregiver.

Implication	Excerpt
Try to look for or assign new meanings in personal activities.	*“I think there was a moment after around 5 years of giving total care that I realized nothing was going to change. That I had to find peace and joy in the life I was living. So, I began to look around for ways to locate purpose and meaning in my life. And I did—these were very little things, but they were meaningful to me.” (Caregiver 001*)
Try to step out of the caregiver role by attending social events, spas, and such.	*“And I manage to do spas three times a year, to maintain my quality of life (...) I go to the music concerts, and to the mass, so that God may bless and protect us.” (Caregiver 020)*
Try to look at caregiving as a dynamic activity that requires continuous learning and adaptation; if possible, attend training.	*“Learning and continuous adaptation. Try to maintain mental and physical balance and thus be possible to balance in the other dimensions of our life (social, family, professional).” (Caregiver 038);* *“I felt prepared, I attended training before, it was an opportunity to be with my mother, which had not happened since I started working.” (Caregiver 052)*
Reach out to healthcare professionals and voice out the concerns.	*“I read a lot, I talked to doctors and nurses. It is difficult, but I can perform with courage and gratitude. I had to leave my job to take care of them.” (Caregiver 020)*

**Table 7 healthcare-09-01337-t007:** Practical implications to the support system.

Support System	Implication	Excerpt
Healthcare professionals	Communicate with family members and clarify who the caregiver is. Make sure the title is given to this person and explain the demands of the role.	*“I did not realize what was happening because nobody tells the caregiver that he became a caregiver... they tell the patient that he is sick... we assume the role and that’s it. As I didn’t have time to prepare, I felt outdated.” (Caregiver 030)*
	Be approachable so that caregivers can reach out and listen to their concerns. Offer support and guidance in the process. Help them prepare for future demands and decisions to be made.	*“Over the years and thanks to the doctors who accompany my mother, I have been able to ‘digest’ or deal better with the whole situation. I am aware that future times will become more difficult and complicated, but I will continue to care and love.” (Caregiver 024);* *“In the beginning, my son had many tantrum attacks. It was a painful process, but with therapies and my participation in training courses on autism, I started to understand my son better.” (Caregiver 005)*
Associations	Continue and increase the range of initiatives to ensure caregivers have a community they can count on. Help employee–caregivers and enterprises communicate and build caregiver-friendly working cultures.	*“We become caregivers—keyword ‘we become’ which presupposes progress. It is not a static process, it’s dynamic, constantly evolving because we must constantly adapt to the progress of illness and the needs of the person being cared for. The association’s support is fantastic in this exercise: we are less adrift!” (Caregiver 030)*
Enterprises	Understand the employee–caregiver need for a flexible schedule and work opportunities. Together, work on concrete solutions to ensure their well-being and proper performance in both and more roles.	*“I changed my whole life. I negotiated schedules with the employer, adapted the house to the best of my ability, and put aside my usual routines. Going out on weekends and socializing with friends has become the last priority, being possible only with the collaboration of third parties.” (Caregiver 064)*
Governments	Work out programs to financially support and facilitate access to home care and house adaptation services. Promote initiatives that help enterprises build caregiver-friendly working cultures. Support associations to increase their initiatives targeting caregivers and enterprises.	*“Outraged by the state’s lack of support (I don’t mean financial) to have time to breathe, rest, sleep and reconcile with employment.” (Caregiver 032);* *“I made an effort, so that they were well and happy, during the day I had the support of a housekeeper.” (Caregiver 020);* *“(I) adapted the house to the best of my ability”. (Caregiver 064)*

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
