# Peer review of "How Does a Family Caregiver’s Sense of Role Loss Impact the Caregiving Experience?"

_healthcare, 2021, doi:10.3390/healthcare9101337_

Round 1
Reviewer 1 Report
Thank you for the opportunity to review this important research. The significant and timely topic deserves to be expressed with precision and clarity. The reader should not be forced to speculate on the authors' intended meaning. I recommend editing to improve readability before submitting for peer review. Examples include the following:
Lines 2-3 - Should either be "How does a family caregiver's sense of role loss" or "How do family caregivers' sense of role . . . "
How does family caregiver’s sense of role loss impact the care-2
giving experience?
Lines 43-47 - unclear what "it" refers to in context of "its impact" and "how to manage it to effectively support" - what does "it" refer to?
Throughout the caring process, family caregivers reportedly have a powerful sense 43
of role loss, felt when one senses a change in role or responsibility, relationship distancing 44
or a changed asymmetry [9]. Little is known about its impact on the caregiving experience 45
and clarifying it will help us understand how caregivers manage it to effectively support 46
them throughout this process.47
Lines 50-54 - The 62-word sentence should be revised for clarity
Our findings shed light on another face of caregiver’s 50
sense of role loss, the sense of caregiver role loss, and point to a possibility of generating 51
it through role rotation and using it as a tool to regain or maintain their sense of personal 52
choice in life and self-priority by allowing caregivers to step out of the role from time to 53
time
Lines 61-62 - meaning unclear
Like others social 61
pattern of behaviors,
Lines 62-65 - 44-word sentence should be revised for clarity
the caregiving role is subjected to norms and rules, it should be seen 62
as a transformation of a current role-relationship, holds strong meaning to incumbents, 63
encompasses expected behaviors under certain social norms or rules dependent on con- 64
text, and any caregiving situation is always unique
Line 88 - should be family caregivers' or the family caregiver's
There is yet a lot to be uncovered about family caregiver’s sense of role loss
Line 94 - insert "the" before family - about the family caregiver's sense - or change caregiver's to caregivers'
There is yet a lot to be uncovered about family caregiver’s sense of role loss
Lines 97-98 - should be caregivers' or the caregiver's
impacting care- 97
giver's well-being a
Lines 126-129 - meaning unclear and lengthy sentences should be revised for clarity
Our findings shed light on another face of caregiver’s sense of role loss, the sense of 126
caregiver role loss, and point to a possibility of generating it through role rotation and 127
using it as a tool to regain or maintain their sense of personal choice in life and self-priority 128
by allowing caregivers to step out of the role from time to time.
Lines 143-144 - meaning unclear
and it consisted if con- 143
sisting of six sections,
Lines 158-159 - meaning unclear
Originally, the research would compare Groups 35 and 27 we believed went through 158
visible role transition and the occurrence of sense of role loss
Table 5 - Impossible to read because no space between columns
Lines 224-225 - Meaning unclear
The first finding is that caregiving intensifies the caregiver’s lived present, and the 224
intensity varies among the groups analyzed, being “time” a recurrent theme in the an- 225
swer
Table 7 - Difficult to read because no space between columns
Table 8 - Difficult to read because no space between columns
Author Response
Ms. Nicole Liu, Managing Editor. Mr. Peter Roth, Publishing Manager.
Healthcare Journal
Healthcare Editorial Office
MDPI, St. Alban-Anlage 66, 4052 Basel, Switzerland
healthcare@mdpi.com
Tel.: +41 61 683 77 34; Fax: +41 61 302 89 18
1st Submission Date: 31 August, 2021. First version.
2nd Submission Date: 23rd September 2021. Revised Version based on Reviewers 1 and 2 Comments and Suggestions (Round 1).
Cover Letter - Reviewer 1
Dear Sir/Madam Reviewer 1:
Thank you for kindly reviewing our article and for concretely pointing out the parts that needed our attention.
We followed your recommendation and re-edited the manuscript to increase the text's precision and clarity, so readers do not need to speculate our intended meaning. Your feedback is very valuable and, after we attended to the points, we feel there was a big improvement.
We hope to have satisfactorily responded to all your comments and suggestions. Please refer to Table 1 for our point-by-point response on next page.
Sincerely,
The authors
Table 1. Authors’ point-by-point response to Reviewer 1’s comments and suggestions.
Revision Part |
Revision Request by Reviewer 1 |
Revision by Authors |
Supplementary Information |
English language and style |
Extensive editing of English language and style required |
· Complete revision of the article language and style to improve clarity. · We rearranged some of the paragraphs and placed them in a more adequate position in the text. The aim was to fully comply with Healthcare Journal’s instructions about what should be presented to readers in each section of the manuscript. · Spell check done. · Alignment of verb tenses for coherence. · Long sentences reduced. Long paragraphs broken into two. |
|
Does the introduction provide sufficient background and include all relevant references? |
Must be improved |
· Language and style revised, spell check done, verb tenses alignment. · Revised to ensure the study was well placed in the broad context so its importance became clear. · Objective and contributions re-written for clarity. · “Theoretical Background” text was fully revised: language and style, paragraph structure for reading fluidity, long sentences reduced to shorter ones. We re-wrote paragraphs with unclear meaning. |
|
Is the research design appropriate? |
Must be improved |
· “Materials and methods” text revised. · Language and style revised to make it clear for reader. · Figure 1 rearranged to be consistent with the description flow. · Table 5’s look and feel improved for better readability: text indentation, justification, cell margins increased. |
|
Are the methods adequately described? |
Must be improved |
· “Materials and methods” text revised. · Language and style revised to make it clear for reader. |
|
Are the conclusions supported by the results? |
Must be improved |
· “Results” text fully revised for clearer presentation. · Language and style revised for readability. Verb tenses aligned for coherence. · Figures 2, 3 and 4 redesigned with larger font size. Text with themes presentation was improved for better readability. |
|
Comments and Suggestions for Authors |
Lines 2-3 - Should either be "How does a family caregiver's sense of role loss" or "How do family caregivers' sense of role . . . "
How does family caregiver’s sense of role loss impact the caregiving experience? |
· We followed the suggestion.
Please refer to the revised text on the right. |
How does a family caregiver’s sense of role loss impact the caregiving experience?
New line(s): 2-3 |
Comments and Suggestions for Authors |
Lines 43-47 - unclear what "it" refers to in context of "its impact" and "how to manage it to effectively support" - what does "it" refer to?
Throughout the caring process, family caregivers reportedly have a powerful sense of role loss, felt when one senses a change in role or responsibility, relationship distancing or a changed asymmetry [9]. Little is known about its impact on the caregiving experience and clarifying it will help us understand how caregivers manage it to effectively support them throughout this process. |
· We re-wrote the text and broke it down into shorter two paragraphs.
Please refer to the revised text on the right. |
Throughout the caring process, family caregivers reportedly have a powerful sense of role loss, felt when one senses a change in role or responsibility, relationship distancing or a changed asymmetry [9] and little is known about how this sense of role loss impacts these individuals' caregiving experiences. To build tailored solutions that will effectively support family caregivers and ac-count for the uniqueness of each caregiving experience [10], it is necessary to develop a deeper understanding of the phenomenon of sense of role loss, its triggers, the impact it had on the caregiver's caregiving experience, how this impact differs among caregiv-ing settings and, from each setting, uncover mechanisms deployed to deal with it.
New Lines: 43-51 |
Comments and Suggestions for Authors |
Lines 50-54 - The 62-word sentence should be revised for clarity
Our findings shed light on another face of caregiver’s sense of role loss, the sense of caregiver role loss, and point to a possibility of generating it through role rotation and using it as a tool to regain or maintain their sense of personal choice in life and self-priority by allowing caregivers to step out of the role from time to time. |
· We re-wrote the text and placed it in a more adequate position in the text.
Please refer to the revised text on the right. |
The objective of this study is to understand the impact of sense of role loss by con-trasting the caregiving experience of family caregivers in different groups where each group configures a different caregiving setting. One of its contributions is bringing to light a new perspective of sense of role loss that can work as a tool with implications on caregiver’s sense of personal choice and self-priority. Furthermore, we researched this phenomenon based on primary role, role transition and contrasted their experi-ences. Prior studies account for caregivers’ experiences based on identity, condition, illness, gender, so we can contribute to this list with these additional settings.
New Lines: 52-59
Our findings shed light on the sense of caregiver role loss and pointed to the possibility of generating it when these individuals rotate through held roles. Role rotation allows family caregivers to step out of the caregiver role from time to time and sense of care-giver role loss can be used as a tool to maintain or regain their sense of personal choice in life and self-priority.
New Lines: 506-510 |
Comments and Suggestions for Authors |
Lines 61-62 - meaning unclear
Like others social pattern of behaviors, |
· We re-wrote the paragraph for clarity.
Please refer to the revised text on the right. |
The family caregiver role emerges from an existing role relationship and should be seen should be seen as this role relationship's transformation, holding strong meaning to those performing in it and regulated by norms or social rules [10]. For instance, a husband-caregiver role can be seen as a transformation of a husband role.
New Lines: 71-74 |
Comments and Suggestions for Authors |
Lines 62-65 - 44-word sentence should be revised for clarity
the caregiving role is subjected to norms and rules, it should be seen as a transformation of a current role-relationship, holds strong meaning to incumbents, encompasses expected behaviors under certain social norms or rules dependent on context, and any caregiving situation is always unique |
· We re-wrote the paragraph for clarity.
Please refer to the revised text on the right. |
The family caregiver role emerges from an existing role relationship and should be seen should be seen as this role relationship's transformation, holding strong meaning to those performing in it and regulated by norms or social rules [10]. For instance, a husband-caregiver role can be seen as a transformation of a husband role.
New Lines: 71-74 |
Comments and Suggestions for Authors |
Line 88 - should be family caregivers' or the family caregiver's
There is yet a lot to be uncovered about family caregiver’s sense of role loss |
· We added “the”.
Please refer to the revised text on the right. |
There is yet a lot to be uncovered about the family caregiver’s sense of role loss
New Line: 99 |
Comments and Suggestions for Authors |
Line 94 - insert "the" before family - about the family caregiver's sense - or change caregiver's to caregivers'
There is yet a lot to be uncovered about family caregiver’s sense of role loss |
· We inserted “the” and revised the paragraph’ text for clarity.
Please refer to the revised text on the right. |
There is yet a lot to be uncovered about the family caregiver’s sense of role loss, its impacts on the caregiving experience, and how these individuals deal with it. In the caregiving context, role changes within the family can occur quickly, may extend for short, long time, or be permanent, resulting in caregivers finding it difficult to adjust to the impact of the care recipient’s illness. Some individuals experience role loss, role gains, taking in the identity or viewing it as an extension of an existing role [25] (p. 122).
New Lines: 99-104 |
Comments and Suggestions for Authors |
Lines 97-98 - should be caregivers' or the caregiver's
impacting caregiver's well-being a |
· We inserted “the” and revised the paragraph’ text for clarity.
Please refer to the revised text on the right. |
Although each caregiving situation is always unique [10], similarities in caregiver role indicate that: (i) majority are female family members performing in it; (ii) the activity changes over time; (iii) caregiving leads to changes in relationships and identities of both caregiver and care recipient; (iv) caregiving is accompanied by stress and burden impacting the caregiver's well-being; (v) and there are positive outcomes such as self-satisfaction and sense of mastery. [26] (pp. 177-182).
New Lines: 105-110 |
Comments and Suggestions for Authors |
Lines 126-129 - meaning unclear and lengthy sentences should be revised for clarity
Our findings shed light on another face of caregiver’s sense of role loss, the sense of caregiver role loss, and point to a possibility of generating it through role rotation and using it as a tool to regain or maintain their sense of personal choice in life and self-priority by allowing caregivers to step out of the role from time to time. |
· Text revised. Paragraphs were placed in a more adequate location in the text.
Please refer to the revised text on the right. |
One of its contributions is bringing to light a new perspective of sense of role loss that can work as a tool with implications on caregiver’s sense of personal choice and self-priority.
New Lines: 52-59
Our findings shed light on the sense of caregiver role loss and pointed to the possibility of generating it when these individuals rotate through held roles. Role rotation allows family caregivers to step out of the caregiver role from time to time and sense of care-giver role loss can be used as a tool to maintain or regain their sense of personal choice in life and self-priority.
New Lines: 506-510 |
Comments and Suggestions for Authors |
Lines 143-144 - meaning unclear
and it consisted if consisting of six sections, |
· We revised the text for clarity.
Please refer to the revised text on the right. |
We employed framework method of qualitative data analysis to organize the data and uncover themes [32] concerning sense of role loss and its impact on the caregiving experience. To create a large and standardized sample of data [33] (p. 361) without causing much inconvenience to potential respondents, empirical data was collected through an online survey with Google Forms [34] (p. 57), as detailed in Table 2.
New Lines: 128-132 |
Comments and Suggestions for Authors |
Lines 158-159 - meaning unclear
Originally, the research would compare Groups 35 and 27 we believed went through visible role transition and the occurrence of sense of role loss |
· We re-wrote the text for clarity.
Please refer to the revised text on the right. |
At first, we would contrast the caregiving experience of Groups 35 and 27 only, because there was a clear before-and-after caregiving activity start. Thus, we assumed members of these groups went through role transition(s) and family caregiver's sense of role loss occurred. But leaving Group 4 out of the analysis would be unfair to its members and a lost opportunity for us to understand these individuals' role transitions, the sense of role loss they felt, and enrich the analysis.
New Lines: 147-152 |
Comments and Suggestions for Authors |
Table 5 - Impossible to read because no space between columns |
· Table’s look and feel improved for better readability: text indentation, justification, cell margins increased. |
|
Comments and Suggestions for Authors |
Lines 224-225 - Meaning unclear
The first finding is that caregiving intensifies the caregiver’s lived present, and the intensity varies among the groups analyzed, being “time” a recurrent theme in the answer |
· We fully revised the text of the “Results” section. We re-wrote this sentence to ensure clarity of what we meant. · Figures were re-designed. · Prior Table 6 and Figure 5 were merged in the new Figure 5, based on Reviewer 2’s suggestion. · Additionally, “Discussion” section was fully revised, too.
Please refer to the revised text on the right. |
The highest score theme in Group 35’s answers reported a need to focus on the present situation accepting one cannot know what comes next (i.e., TH063). In Group 27, similar theme ranked third and it was phrased as a need to be constantly alert, monitor the situation, and adapt to it (i.e., TH019), but did not appear in the answers of Group 4. So, the first finding of this study was that caregiving requires that caregivers have a heightened focus on the present circumstance to respond fast to the intense, constantly changing demands of caregiving and this varied among the three settings. Family caregivers find themselves devoting different amounts of their time to support care recipients the best way they can. Time, or the lack of it, was a recurrent theme in their answers.
New Lines: 217-225 |
Comments and Suggestions for Authors |
Table 7 - Difficult to read because no space between columns |
· Table’s look and feel improved for better readability: text indentation, justification, cell margins increased. · Text was revised for clarity. |
Tables 7 and 8 are now Tables 6 and 7, respectively. Prior Table 6 was deleted after Reviewer 2’s suggestion. |
Comments and Suggestions for Authors |
Table 8 - Difficult to read because no space between columns |
· Table’s look and feel improved for better readability: text indentation, justification, cell margins increased. · Text was revised for clarity. |
Tables 7 and 8 are now Tables 6 and 7, respectively. Prior Table 6 was deleted after Reviewer 2’s suggestion. |

Reviewer 2 Report
The authors conducted a study to analyze the relationship between family caregiver's sense of role loss and the caregiving experience. The introduction is informative with background of the study. The methods are described clearly with details. A few suggestions which may further improve the manuscript.
- Figure quality could be improved. Most figures have low resolution in the manuscript.
- Table. 6 and Fig. 5 are not informative. Maybe the authors could consider changing them to be text description.
- The excerpt parts seem too long to put into tables. e.g. Table. 7 and 8.
Author Response
Ms. Nicole Liu, Managing Editor. Mr. Peter Roth, Publishing Manager.
Healthcare Journal
Healthcare Editorial Office
MDPI, St. Alban-Anlage 66, 4052 Basel, Switzerland
healthcare@mdpi.com
Tel.: +41 61 683 77 34; Fax: +41 61 302 89 18
1st Submission Date: 31 August, 2021. First version.
2nd Submission Date: 23rd September 2021. Revised Version based on Reviewers 1 and 2 Comments and Suggestions (Round 1).
Cover Letter - Reviewer 2
Dear Sir/Madam Reviewer 2:
Thank you for kindly reviewing our article and for concretely pointing out the parts that needed our attention.
We greatly appreciate the suggestions proposed and, after we attended to them, we feel there is clear improvement to the manuscript.
We hope to have satisfactorily responded to all your comments and suggestions. Please refer to Table 1 for our point-by-point response on next page.
Sincerely,
The authors
Table 1. Authors’ point-by-point response to Reviewer 2’s comments and suggestions.
Revision Part |
Revision Request by Reviewer 2 |
Revision by Authors |
Supplementary Information |
English language and style |
English language and style are fine/minor spell check required |
· Complete revision of the article language and style to improve clarity. · Spell check done. · Alignment of verb tenses for coherence. |
|
Are the methods adequately described? |
Can be improved |
· “Materials and methods” text revised. · Language and style revised to make it clear for reader. · Figure 1 rearranged to be consistent with the description flow. · Table 5’s look and feel improved for better readability: text indentation, justification, cell margins increased. |
|
Are the results clearly presented? |
Can be improved |
· “Results” text fully revised for clearer presentation. Additionally, “Discussion” section was revised, too. · Language and style revised for readability. · Verb tenses aligned for coherence. · Figures 2, 3 and 4 redesigned with larger font size. Text with themes presentation was improved for better readability. |
|
Comments and Suggestions for Authors |
Figure quality could be improved. Most figures have low resolution in the manuscript. |
· All figures had their resolution increased to 330dpi. Please refer to image on the right side. |
|
Comments and Suggestions for Authors |
Table. 6 and Fig. 5 are not informative. Maybe the authors could consider changing them to be text description |
· We merged Tab. 6 and Fig. 5 in one Figure 5. Please see image on the right. · The text description was revised for clarity and to fit the new figure. |
|
Comments and Suggestions for Authors |
The excerpt parts seem too long to put into tables. e.g. Table. 7 and 8. |
· The excerpts were re-checked to see if parts of them could be removed, but we felt that the change to the family caregivers’ testimonies would not entirely convey their message. We thought it was important to reserve this space so we could hear them. Thus, we revised the tables’ look and feel for improved readability (text indentation, justification and cell margins increased). |
Tables 7 and 8 are now Tables 6 and 7, respectively. |

Round 2
Reviewer 1 Report
Thank you for incorporating recommendations and highlighting the changes to facilitate review. The text revisions are excellent. Graphics would benefit from additional space between columns; text should be block style, rather than indented beneath the first line. The goal is to create open space/white space between data to improve readability. The substantive study results deserve to be presented with precision and clarity to help readers understand data.
Table 4 is an example of clear presentation.
The Table 4 style should be used in the following:
- Tables 1, 2, 3, 5, 6, 7
- Figures 2, 3, 4
- Appendix A
Thank you.
Author Response
Ms. Nicole Liu, Managing Editor. Mr. Peter Roth, Publishing Manager.
Healthcare Journal
Healthcare Editorial Office
MDPI, St. Alban-Anlage 66, 4052 Basel, Switzerland
healthcare@mdpi.com
Tel.: +41 61 683 77 34; Fax: +41 61 302 89 18
1st Submission Date: 31 August, 2021. First version.
2nd Submission Date: 23rd September 2021. Revised Version based on Reviewers 1 and 2 Comments and Suggestions (Round 1).
3rd Submission Date: 27th September 2021. Revised Version based on Reviewer 1’s Comments and Suggestions (Round 2).
Cover Letter - Reviewer 1
Dear Sir/Madam Reviewer 1:
We are happy that you were satisfied with the changes to the manuscript. Thank you again for taking the time to review the round 1’s revised version and kindly suggesting a better layout for the tables and figures.
We followed your recommendation and applied Table 4’s layout to the others. Figures were revised, too. Based on your suggestions, we:
Tables 1, 2, 3, 5, 6, 7, Appendix A |
Figures 2, 3, 4 |
We reverted the text to block style |
Increased the width of the rectangles/squares to avoid line break between grouped themes |
Increased the space between the columns |
Reverted the text with themes to block style |
Added a border between the rows
|
Increased the spaces between the text with themes |
After we implemented the changes to the layouts, we feel it there was a big improvement again. They are highlighted in green colour.
Again, we hope to have satisfactorily responded to your suggestions and were able to increase the precision and clarity of the results so readers can easily understand the data.
Sincerely,
The authors
